# Exact Learning Dynamics of Bottlenecked and Wide Deep Linear Networks

**Clémentine C J Dominé**
IST Austria
clementine.domine98@gmail.com

## Abstract

Architectural diversity shapes learning in both biological and artificial systems, influencing how features are represented and generalized. Motivated by parallels between brain circuitry and machine learning architectures, we study the exact learning dynamics of wide and narrow two-layer linear networks, extending the class of solvable models to encompass a broader range of architectural configurations. Our framework captures how depth, width, and bottleneck structures affect feature learning. This approach has the potential to advance theoretical understanding of network architectures relevant to neuroscience—such as cerebellum-like structures—but also informs practical designs in modern machine learning, including in Low-Rank Adaptation (LoRA) modules.

## 1   Introduction

Architectural diversity is a hallmark of both biological and artificial learning systems. In neuroscience, distinct brain regions exhibit a variety of structural organizations, with differences in connectomics thought to reflect the diverse functional roles these areas serve Lappalainen et al. [2024]. Similarly, in machine learning, parameters such as network depth, width, and connectivity are systematically varied to assess their influence on learning dynamics, structured representations, and overall performance Dominé et al. [2024], Kunin et al. [2024a]. Extending solvable models to better capture real-world complexities is crucial in both domains, not only for advancing functional understanding but also for guiding practical applications. This is particularly relevant for architectures with established connections to neuroscience, such as deep feed-forward networks, wide networks, and bottleneck designs, which have been used to model brain regions including cerebellum-like structures Muscinelli et al. [2023], Nogueira et al. [2023]. These same architectural motifs also play a significant role in modern machine learning practice. For example, Low-Rank Adaptation (LoRA) modules Hu et al. [2021] have emerged as a highly effective method for fine-tuning large pre-trained models in natural language processing and computer vision Hu et al. [2021], Dettmers et al. [2023]. In this work, we derive the exact dynamics of both wide and narrow networks and investigate how these architectural choices shape feature learning. By expanding the set of solvable models and analyzing their relation to feature learning, we aim to enhance the characterization, interpretability, and performance of both machine learning architectures and theoretical models of brain function.

## 2   Related Work

**Linear networks.** Our study builds on an extensive literature on deep linear networks, which, despite their apparent simplicity, have proven to be powerful tools for understanding the behavior of more complex neural architectures [Baldi and Hornik, 1989, Fukumizu, 1998, Saxe et al., 2014]. Prior work has examined their convergence properties [Arora et al., 2018a, Du and Hu, 2019], generalization behavior [Lampinen and Ganguli, 2018, Poggio et al., 2018, Huh, 2020], and the implicit bias

introduced by gradient descent [Arora et al., 2019a, Woodworth et al., 2020, Chizat and Bach, 2020, Kunin et al., 2022]. These studies have revealed that deep linear networks exhibit rich fixed-point structures and non-linear learning dynamics in both parameter and function space—features reminiscent of their non-linear counterparts [Arora et al., 2018b, Lampinen and Ganguli, 2018]. Seminal work by Saxe et al. [2014] provided exact solutions to gradient flow dynamics under task-aligned initializations, showing that the largest singular modes are learned first during training. Subsequent research extended this analysis to broader classes of deep linear networks Arora et al. [2018b, 2019a], Ziyin et al. [2022] and developed more flexible initialization schemes [Gidel et al., 2019, Tarmoun et al., 2021, Gissin et al., 2019]. Our work builds directly on the matrix Riccati framework introduced in Fukumizu [1998], Braun et al. [2022b] and extended in Dominé et al. [2024], which derived dynamics for wide $\lambda$-balanced networks and clarified how initialization influences both the *rich* and *lazy* learning regimes. Here, we extend these solutions to bottlenecked and wide architectures, making the solution applicable to any type of feedforward architecture.

**Architecture.** Analyses of learning dynamics often rely on stringent dimensionality constraints, which can limit the scope and generality of their conclusions. A substantial body of work has examined how initialization variance and layer-wise learning rates must scale in the infinite-width limit to preserve stable activations, gradients, and outputs. In contrast, our work derives explicit analytical solutions for both network dynamics and the associated NTK in settings that move beyond the infinite-width idealization—covering large but finite widths as well as finite-width bottleneck architectures [Jacot et al., 2021, Xu and Ziyin, 2024, Kunin et al., 2024b, Chizat et al., 2019]. Earlier studies in finite-width regimes [Fukumizu, 1998, Braun et al., 2022b, Dominé et al., 2024, Kunin et al., 2024b, Xu and Ziyin, 2024] imposed restrictive dimensionality assumptions; here, we revisit and relax these constraints, thereby broadening the range of architectures for which exact solutions can be obtained. Our analysis is confined to the two-layer setting and does not yet extend to deeper networks. While the role of depth in shaping learning dynamics has been extensively investigated Saxe et al. [2014], such studies typically assume that the network's singular structure is aligned with that of the target task. By contrast, our framework achieves exact solutions without requiring this alignment, offering a more general account of architectural effects on learning.

## 3    Preliminaries

Consider a supervised learning task where input vectors $\mathbf{x}_n \in \mathbb{R}^{N_i}$, from a set of $P$ training pairs $\{(\mathbf{x}_n, \mathbf{y}_n)\}_{n=1}^P$, need to be mapped to their corresponding target output vectors $\mathbf{y}_n \in \mathbb{R}^{N_o}$. We learn this task with a two-layer linear network model that produces the output prediction $\hat{\mathbf{y}}_n = \mathbf{W}_2 \mathbf{W}_1 \mathbf{x}_n$, with weight matrices $\mathbf{W}_1 \in \mathbb{R}^{N_h \times N_i}$ and $\mathbf{W}_2 \in \mathbb{R}^{N_o \times N_h}$, where $N_h$ is the number of hidden units. The network's weights are optimized using full batch gradient descent with learning rate $\eta$ (or respectively time constant $\tau = \frac{1}{\eta}$) on the mean squared error loss $\mathcal{L}(\hat{\mathbf{y}}, \mathbf{y}) = \frac{1}{2} \langle ||\hat{\mathbf{y}} - \mathbf{y}||^2 \rangle$, where $\langle \cdot \rangle$ denotes the average over the dataset. We employ an approach first introduced in the foundational work of Fukumizu [1998] and extended in recent work by Braun et al. [2022b], which instead of studying the parameters directly, considers the dynamics of a matrix of important statistics. In particular, defining $\mathbf{Q} = \begin{bmatrix} \mathbf{W}_1 & \mathbf{W}_2^T \end{bmatrix}^T \in \mathbb{R}^{(N_i + N_o) \times N_h}$, we consider the $(N_i + N_o) \times (N_i + N_o)$ matrix

$$\mathbf{Q}\mathbf{Q}^T(t) = \begin{bmatrix} \mathbf{W}_1^T \mathbf{W}_1(t) & \mathbf{W}_1^T \mathbf{W}_2^T(t) \\ \mathbf{W}_2 \mathbf{W}_1(t) & \mathbf{W}_2 \mathbf{W}_2^T(t) \end{bmatrix}, \tag{1}$$

which is divided into four quadrants with interpretable meanings, and where $t \in \mathbb{R}$ represents training time. The approach monitors several key statistics collected in the matrix. The off-diagonal blocks contain the network function $\hat{\mathbf{Y}}(t) = \mathbf{W}_2 \mathbf{W}_1(t)\mathbf{X}$. The on-diagonal blocks capture the correlation structure of the weight matrices, allowing for the calculation of the temporal evolution of the network's internal representations. This includes the representational similarity matrices (RSM) of the neural representations within the hidden layer, as first defined by Braun et al. [2022b], $\text{RSM}_I = \mathbf{X}^T \mathbf{W}_1^T \mathbf{W}_1(t)\mathbf{X}$, $\text{RSM}_O = \mathbf{Y}^T (\mathbf{W}_2 \mathbf{W}_2^T(t))^+ \mathbf{Y}$, where $+$ denotes the pseudoinverse; and the network's finite-width NTK [Jacot et al., 2018, Lee et al., 2019, Arora et al., 2019b] $\text{NTK} = \mathbf{I}_{N_o} \otimes \mathbf{X}^T \mathbf{W}_1^T \mathbf{W}_1(t)\mathbf{X} + \mathbf{W}_2 \mathbf{W}_2^T(t) \otimes \mathbf{X}^T \mathbf{X}$, where $\mathbf{I}_{N_o}$ is the $N_o \times N_o$ identity matrix and $\otimes$ is the Kronecker product. Hence, the dynamics of $\mathbf{Q}\mathbf{Q}^T$ describe the important aspects of network behaviour.

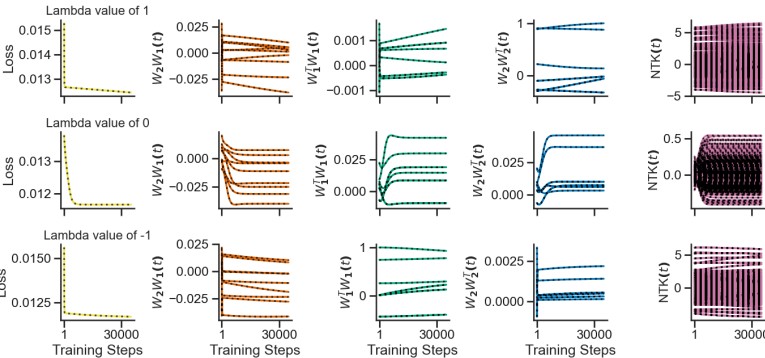

Figure 1: **bottlenecked Network Dynamics A.** The temporal dynamics of the numerical simulation (colored lines) of the loss, network function, correlation of input and output weights, and the NTK are exactly matched by the analytical solution (black dotted lines) for $\lambda = -1$. **B.** $\lambda = 0$. **C.** $\lambda = -1$ initial weight values initialized as described in AppendixB.

## 4 Exact Learning Dynamics

We derive an exact solution for $\mathbf{Q}\mathbf{Q}^T$ offering insight into the learning dynamics, convergence behavior, and generalization properties of two-layer linear networks with prior knowledge.

**Assumptions.** To derive these solutions we make the following assumptions:

- **A1** (*Whitened input*). The input data is whitened, i.e. $\tilde{\mathbf{\Sigma}}^{xx} = \mathbf{I}$.
- **A2** ($\lambda$-*Balanced*). The network's weight matrices are $\lambda$-balanced at the beginning of training, i.e. $\mathbf{W}_2^T\mathbf{W}_2(0) - \mathbf{W}_1\mathbf{W}_1(0)^T = \lambda\mathbf{I_h}$. If this condition holds at initialization, it will persist throughout training [Saxe et al., 2014, Arora et al., 2018a].

Our assumptions are weaker than those in prior work [Fukumizu, 1998, Braun et al., 2022b, Kunin et al., 2024b, Xu and Ziyin, 2024, Dominé et al., 2024], which imposed stronger dimensionality constraints. For example, Fukumizu [1998] required equal input and output dimensions ($N_i = N_o$) with $\lambda = 0$, while Braun et al. [2022a], Dominé et al. [2024] allowed $N_i \neq N_o$ and arbitrary $\lambda$, but restricted attention to non-bottleneck networks ($N_h = \min(N_i, N_o)$) and non-wide networks. In contrast, we further relax these conditions by considering arbitrary hidden-layer widths, including bottlenecked architectures, with general values of $\lambda$ at initialization. It is worth noting that the relaxation to wide networks provides only limited benefits: the effective rank of the model remains bounded regardless of the hidden layer width, so expressivity does not improve. Nevertheless, narrow and bottlenecked networks remain highly relevant with established connections to neuroscience.

Within this broader setting, we are able to compute the exact temporal dynamics of the loss, network function, representational similarity matrices (RSMs), and NTK (Fig.1) across a range of $\lambda$-balanced initializations of bottlenecked networks. The full statement and proof of the theorem on exact learning dynamics are provided in AppendixA. Our derivation follows the approach of Dominé et al. [2024], but with two key differences. First, we explicitly specify the dimensionality of the identity matrix in the solutions as $N_h$ and $N_{\min(N_i,N_o)}$, whereas earlier works implicitly constrained all of them $N_h = \min(N_i, N_o)$. Second, we introduce a novel algorithm for constructing $\lambda$-balanced initializations in bottlenecked networks. This generalization reveals a wide spectrum of learning dynamics across different values of $\lambda$. Notably, bottlenecked networks never achieve zero loss as expected.

## 5 Application: Rich and Lazy Learning in Bottlenecked Networks

Previous work by Dominé et al. [2024] showed that the interaction between the magnitude of the *relative scale* parameter $\lambda$ and the network architecture determines the balance between rich and

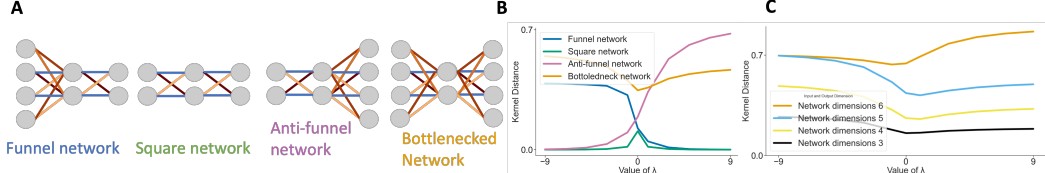

Figure 2: **A.** Schematic representations of the network architectures considered, from left to right: funnel network, square network, inverted-funnel network, and bottleneck network. **B.** The plot shows the NTK kernel distance from initialization, as defined in Fort et al. [2020] across the four architectures depicted schematically. **C.** The NTK kernel distance away from initialization as a function of bottleneck proportion.

lazy learning regimes. Building on these findings, we extend the analysis to bottlenecked networks. Specifically, we examine four network architectures, illustrated in Fig. 2A

1. *Funnel networks* — narrowing from input to output ($N_i > N_h = N_o$),
2. *Inverted-funnel networks* — expanding from input to output ($N_i = N_h < N_o$),
3. *Square networks* — equal input, hidden, and output dimensions ($N_i = N_h = N_o$), and
4. *Bottlenecked networks* — with $N_h < \min(N_i, N_o)$.

Our solution, $\mathbf{QQ}^T$, captures the NTK dynamics across all these architectures. To study how the NTK evolves under different $\lambda$ initializations, we compute the kernel distance from initialization following Fort et al. [2020]. As shown in Fig. 2B, our results reproduce the findings of Dominé et al. [2024]: funnel networks enter the *lazy* regime as $\lambda \to \infty$, while inverted-funnel networks do so as $\lambda \to -\infty$. Conversely, in the opposite $\lambda$ limits, these architectures transition from the *lazy* to the *rich* regime. For square networks, as $\lambda \to \pm\infty$, the network consistently enters the lazy regime, whereas across all architectures, as $\lambda \to 0$, the dynamics shift into the rich regime. Interestingly, bottlenecked networks remain in the rich regime regardless of $\lambda$. This can be attributed to their limited capacity to perfectly fit the target function. Intuitively, the networks can not learn the function in the lazy regime, since the rank of the network is smaller than that of the task. This phenomenon is consistently observed across different architectural families as shown in Fig. 2C.

## 6 Conclusion

In conclusion, we have derived the dynamics of wide and narrow two layer linear networks, examining how architectural choices influence feature learning. This framework enables a detailed analysis of representational dynamics throughout the learning process. A limitation of our approach is that, while it relaxes many of the architectural constraints, it does not directly extend to deep network architectures, we leave this for future work. Looking forward, we also plan to extend this approach to the study of LoRA architectures, with the aim of providing a theoretical foundation for their empirical success and identifying avenues for further improvement. Additionally, we intend to conduct a systematic analysis of linear autoencoders, with particular emphasis on sparse auto-encoders—currently employed to extract interpretable features from large language models Cunningham et al. [2023]. We also plan to further investigate how architectural choices influence performance in complex learning scenarios, including transfer learning, continual learning, and reversal learning.

## 7 Acknowledgement

C.D contribution to this research was funded in part by the Austrian Science Fund (FWF) 10.55776/COE12.

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

# A    Exact Learning Dynamics

## A.1   $\mathbf{Q}\mathbf{Q}^T$ diagonalization

**Lemma A.1.** *If $\mathbf{F} = \mathbf{P}\mathbf{\Lambda}\mathbf{P}^T$ is symmetric and diagonalisable, then the matrix Riccati differential equation $\frac{d}{dt}(\mathbf{Q}\mathbf{Q}^T) = \mathbf{F}\mathbf{Q}\mathbf{Q}^T + \mathbf{Q}\mathbf{Q}^T\mathbf{F} - (\mathbf{Q}\mathbf{Q}^T)^2$ with initialisation $\mathbf{Q}\mathbf{Q}^T(0) = \mathbf{Q}(0)\mathbf{Q}(0)^T$ has a unique solution for all $t \geq 0$, and the solution is given by*

$$\mathbf{Q}\mathbf{Q}^T(t) = e^{\mathbf{F}\frac{t}{\tau}}\mathbf{Q}(0)\left[+\mathbf{I_h} + \mathbf{Q}(0)^T\mathbf{P}\left(\frac{e^{2\mathbf{\Lambda}\frac{t}{\tau}} - \mathbf{I}_{\min(\mathbf{N_i},\mathbf{N_o})}}{2\mathbf{\Lambda}}\right)\mathbf{P}^T\mathbf{Q}(0)\right]^{-1}\mathbf{Q}(0)^T e^{\mathbf{F}\frac{t}{\tau}}. \quad (2)$$

*This is true even when there exists $\mathbf{\Lambda}_i = 0$.*

*Proof.* First we show that there exists a unique solution to the initial value problem stated. This is true by Picard-Lindelöf theorem. Now we show that the provided solution satisfies the ODE. Let $\mathbf{L} = e^{\mathbf{F}\frac{t}{\tau}}\mathbf{Q}(0)$ and $\mathbf{C} = \mathbf{I} + \mathbf{Q}(0)^T\mathbf{P}\left(\frac{e^{2\mathbf{\Lambda}\frac{t}{\tau}} - \mathbf{I}_{\min(\mathbf{N_i},\mathbf{N_o})}}{2\mathbf{\Lambda}}\right)\mathbf{P}^T\mathbf{Q}(0)$ such that solution $\mathbf{Q}\mathbf{Q}^T(t) = \mathbf{L}\mathbf{C}^{-1}\mathbf{L}^T$. The time derivative of $\mathbf{Q}\mathbf{Q}^T$ is then given by

$$\frac{d}{dt}(\mathbf{Q}\mathbf{Q}^T) = \frac{d}{dt}(\mathbf{L})\mathbf{C}^{-1}\mathbf{L}^T + \mathbf{L}\frac{d}{dt}(\mathbf{C}^{-1})\mathbf{L}^T + \mathbf{L}\mathbf{C}^{-1}\frac{d}{dt}(\mathbf{L}^T). \quad (3)$$

Solving for these derivatives individually, we find

$$\frac{d}{dt}(\mathbf{L}) = \frac{d}{dt}e^{\mathbf{F}\frac{t}{\tau}}\mathbf{Q}(0) = \mathbf{F}e^{\mathbf{F}\frac{t}{\tau}}\mathbf{Q}(0) = \mathbf{F}\mathbf{L}, \quad (4)$$

$$\frac{d}{dt}(\mathbf{C}^{-1}) = -\mathbf{C}^{-1}\frac{d}{dt}(\mathbf{C})\mathbf{C}^{-1} = -\mathbf{C}^{-1}\mathbf{Q}(0)^T\mathbf{P}\frac{d}{dt}\left(\frac{e^{2\mathbf{\Lambda}\frac{t}{\tau}} - \mathbf{I}_{\min(\mathbf{N_i},\mathbf{N_o})}}{2\mathbf{\Lambda}}\right)\mathbf{P}^T\mathbf{Q}(0)\mathbf{C}^{-1}. \quad (5)$$

We consider the derivative of the fraction separately,

$$\frac{d}{dt}\left(\frac{e^{2\mathbf{\Lambda}\frac{t}{\tau}} - \mathbf{I}_{\min(\mathbf{N_i},\mathbf{N_o})}}{2\mathbf{\Lambda}}\right) = e^{2\mathbf{\Lambda}\frac{t}{\tau}}, \quad (6)$$

this is true even in the limit as $\lambda_i \to 0$. Plugging these derivatives back in we see that the solution satisfies the ODE. Lastly, let $t = 0$, we see that the the solution satisfies the initial conditions. $\square$

## A.2   F   diagonalisation

**Lemma A.2.** *The eigendecomposition of $\mathbf{F} = \mathbf{P}\mathbf{\Lambda}\mathbf{P}^T$ where*

$$\mathbf{P} = \frac{1}{\sqrt{2}}\begin{pmatrix} \tilde{\mathbf{V}}(\tilde{\mathbf{G}} - \tilde{\mathbf{H}}\tilde{\mathbf{G}}) & \tilde{\mathbf{V}}(\tilde{\mathbf{G}} + \tilde{\mathbf{H}}\tilde{\mathbf{G}}) & \sqrt{2}\tilde{\mathbf{V}}_\perp \\ \tilde{\mathbf{U}}(\tilde{\mathbf{G}} + \tilde{\mathbf{H}}\tilde{\mathbf{G}}) & -\tilde{\mathbf{U}}(\tilde{\mathbf{G}} - \tilde{\mathbf{H}}\tilde{\mathbf{G}}) & \sqrt{2}\tilde{\mathbf{U}}_\perp \end{pmatrix}, \quad \mathbf{\Lambda} = \begin{pmatrix} \tilde{\mathbf{S}}_\lambda & 0 & 0 \\ 0 & -\tilde{\mathbf{S}}_\lambda & 0 \\ 0 & 0 & \boldsymbol{\lambda}_\perp \end{pmatrix} \quad (7)$$

*and the matrices $\tilde{\mathbf{S}}_\lambda$, $\boldsymbol{\lambda}_\perp$, $\tilde{\mathbf{H}}$, and $\tilde{\mathbf{G}}$ are the diagonal matrices defined as:*

$$\tilde{\mathbf{S}}_\lambda = \sqrt{\tilde{\mathbf{S}}^2 + \frac{\lambda^2}{4}\mathbf{I}_{\min(\mathbf{N_i},\mathbf{N_o})}}, \quad \boldsymbol{\lambda}_\perp = \mathrm{sgn}(|N_o - N_i|)\frac{\lambda}{2}\mathbf{I}_{\mathbf{N_o}-\mathbf{N_i}}, \quad \tilde{\mathbf{H}} = \mathrm{sgn}(\lambda)\sqrt{\frac{\tilde{\mathbf{S}}_\lambda - \tilde{\mathbf{S}}}{\tilde{\mathbf{S}}_\lambda + \tilde{\mathbf{S}}}}, \quad \tilde{\mathbf{G}} = \frac{1}{\sqrt{\mathbf{I}_{\min(\mathbf{N_i},\mathbf{N_o})} + \tilde{\mathbf{H}}^2}}.$$
$$\quad (8)$$

Beyond the invertibility of $F$, we need to understand the relationship between $F$ and $Q(0)$. To do this the following lemma relates the structure between the SVD of the model with the SVD structure of the individual parameters.

*Proof.* We leave for the reader by computing

$$\mathbf{F} = \mathbf{P}\mathbf{\Lambda}\mathbf{P}^T. \quad (9)$$

$\square$

## A.3 Solution unequal-input-output

**Theorem A.3.** *Under the assumptions of whitened inputs (Assumption 1), lambda-balanced weights (Assumption 2),the temporal dynamics of $\mathbf{QQ}^T$ are*

$$\mathbf{QQ}^T(t) = \begin{pmatrix} \mathbf{Z}_1\mathbf{A}^{-1}\mathbf{Z}_1^T & \mathbf{Z}_1\mathbf{A}^{-1}\mathbf{Z}_2^T \\ \mathbf{Z}_2\mathbf{A}^{-1}\mathbf{Z}_1^T & \mathbf{Z}_2\mathbf{A}^{-1}\mathbf{Z}_2^T \end{pmatrix}, \tag{10}$$

*where the variables $\mathbf{Z}_1 \in \mathbb{R}^{N_i \times N_h}$, $\mathbf{Z}_2 \in \mathbb{R}^{N_o \times N_h}$, and $\mathbf{A} \in \mathbb{R}^{N_h \times N_h}$ are defined as*

$$\mathbf{Z}_1(t) = \frac{1}{2}\tilde{\mathbf{V}}(\tilde{\mathbf{G}} - \tilde{\mathbf{H}}\tilde{\mathbf{G}})e^{\tilde{\mathbf{S}}_\lambda \frac{t}{\tau}}\mathbf{B}^T - \frac{1}{2}\tilde{\mathbf{V}}(\tilde{\mathbf{G}} + \tilde{\mathbf{H}}\tilde{\mathbf{G}})e^{-\tilde{\mathbf{S}}_\lambda \frac{t}{\tau}}\mathbf{C}^T + \tilde{\mathbf{V}}_\perp e^{\lambda_\perp \frac{t}{\tau}}\tilde{\mathbf{V}}_\perp^T\mathbf{W}_1(0)^T, \tag{11}$$

$$\mathbf{Z}_2(t) = \frac{1}{2}\tilde{\mathbf{U}}(\tilde{\mathbf{G}} + \tilde{\mathbf{H}}\tilde{\mathbf{G}})e^{\tilde{\mathbf{S}}_\lambda \frac{t}{\tau}}\mathbf{B}^T + \frac{1}{2}\tilde{\mathbf{U}}(\tilde{\mathbf{G}} - \tilde{\mathbf{H}}\tilde{\mathbf{G}})e^{-\tilde{\mathbf{S}}_\lambda \frac{t}{\tau}}\mathbf{C}^T + \tilde{\mathbf{U}}_\perp e^{\lambda_\perp \frac{t}{\tau}}\tilde{\mathbf{U}}_\perp^T\mathbf{W}_2(0), \tag{12}$$

$$\mathbf{A}(t) = \mathbf{I_h} + \mathbf{B}\left(\frac{e^{2\tilde{\mathbf{S}}_\lambda \frac{t}{\tau}} - \mathbf{I_{min(N_i,N_o)}}}{4\tilde{\mathbf{S}}_\lambda}\right)\mathbf{B}^T - \mathbf{C}\left(\frac{e^{-2\tilde{\mathbf{S}}_\lambda \frac{t}{\tau}} - \mathbf{I_{min(N_i,N_o)}}}{4\tilde{\mathbf{S}}_\lambda}\right)\mathbf{C}^T \tag{13}$$

$$+ \mathbf{W}_2(0)^T\tilde{\mathbf{U}}_\perp\left(\frac{e^{\lambda_\perp \frac{t}{\tau}} - \mathbf{I_{min(N_i,N_o)}}}{\lambda_\perp}\right)\tilde{\mathbf{U}}_\perp^T\mathbf{W}_2(0) + \mathbf{W}_1(0)\tilde{\mathbf{V}}_\perp\left(\frac{e^{\lambda_\perp \frac{t}{\tau}} - \mathbf{I_{min(N_i,N_o)}}}{\lambda_\perp}\right)\tilde{\mathbf{V}}_\perp^T\mathbf{W}_1(0)^T.$$

*Proof.* We start and use the diagonalisation of $\mathbf{F}$ to rewrite the matrix exponential of $\mathbf{F}$ and $\mathbf{F}$. Note that $\mathbf{P}^T\mathbf{P} = \mathbf{PP}^T = \mathbf{I}$ and therefore $\mathbf{P}^T = \mathbf{P}^{-1}$.

$$e^{\mathbf{F}\frac{t}{\tau}} = \mathbf{P}e^{\mathbf{\Gamma}}\mathbf{P}^\intercal$$

$$= \frac{1}{\sqrt{2}}\begin{bmatrix} \tilde{\mathbf{V}}(\tilde{\mathbf{G}} - \tilde{\mathbf{H}}\tilde{\mathbf{G}}) & \tilde{\mathbf{V}}(\tilde{\mathbf{G}} + \tilde{\mathbf{H}}\tilde{\mathbf{G}}) & \sqrt{2}\mathbf{V}_\perp \\ \tilde{\mathbf{U}}(\tilde{\mathbf{G}} + \tilde{\mathbf{H}}\tilde{\mathbf{G}}) & -\tilde{\mathbf{U}}(\tilde{\mathbf{G}} - \tilde{\mathbf{H}}\tilde{\mathbf{G}}) & \sqrt{2}\mathbf{U}_\perp \end{bmatrix}\begin{bmatrix} e^{\tilde{\mathbf{S}}_\lambda \frac{t}{\tau}} & 0 & 0 \\ 0 & e^{-\tilde{\mathbf{S}}_\lambda \frac{t}{\tau}} & 0 \\ 0 & 0 & e^{\lambda_\perp \frac{t}{\tau}} \end{bmatrix}$$

$$\times \frac{1}{\sqrt{2}}\begin{bmatrix} \tilde{\mathbf{V}}(\tilde{\mathbf{G}} - \tilde{\mathbf{H}}\tilde{\mathbf{G}}) & \tilde{\mathbf{V}}(\tilde{\mathbf{G}} + \tilde{\mathbf{H}}\tilde{\mathbf{G}}) & \sqrt{2}\mathbf{V}_\perp \\ \tilde{\mathbf{U}}(\tilde{\mathbf{G}} + \tilde{\mathbf{H}}\tilde{\mathbf{G}}) & -\tilde{\mathbf{U}}(\tilde{\mathbf{G}} - \tilde{\mathbf{H}}\tilde{\mathbf{G}}) & \sqrt{2}\mathbf{U}_\perp \end{bmatrix}^T$$

$$= \frac{1}{\sqrt{2}}\begin{bmatrix} \tilde{\mathbf{V}}(\tilde{\mathbf{G}} - \tilde{\mathbf{H}}\tilde{\mathbf{G}}) & \tilde{\mathbf{V}}(\tilde{\mathbf{G}} + \tilde{\mathbf{H}}\tilde{\mathbf{G}}) \\ \tilde{\mathbf{U}}(\tilde{\mathbf{G}} - \tilde{\mathbf{H}}\tilde{\mathbf{G}}) & -\tilde{\mathbf{U}}(\tilde{\mathbf{G}} + \tilde{\mathbf{H}}\tilde{\mathbf{G}}) \end{bmatrix}\begin{bmatrix} e^{\tilde{\mathbf{S}}_\lambda \frac{t}{\tau}} & 0 \\ 0 & e^{-\tilde{\mathbf{S}}_\lambda \frac{t}{\tau}} \end{bmatrix}$$

$$\times \frac{1}{\sqrt{2}}\begin{bmatrix} \tilde{\mathbf{V}}(\tilde{\mathbf{G}} - \tilde{\mathbf{H}}\tilde{\mathbf{G}}) & \tilde{\mathbf{V}}(\tilde{\mathbf{G}} + \tilde{\mathbf{H}}\tilde{\mathbf{G}}) \\ \tilde{\mathbf{U}}(\tilde{\mathbf{G}} + \tilde{\mathbf{H}}\tilde{\mathbf{G}}) & -\tilde{\mathbf{U}}(\tilde{\mathbf{G}} - \tilde{\mathbf{H}}\tilde{\mathbf{G}}) \end{bmatrix}^T$$

$$\tag{14}$$

$$+ 2\frac{1}{\sqrt{2}}\begin{bmatrix} \tilde{\mathbf{V}}_\perp \\ \tilde{\mathbf{U}}_\perp \end{bmatrix}e^{\lambda_\perp \frac{t}{\tau}}\frac{1}{\sqrt{2}}\begin{bmatrix} \tilde{\mathbf{V}}_\perp \\ \tilde{\mathbf{U}}_\perp \end{bmatrix}^T$$

$$= \mathbf{O}e^{\mathbf{\Lambda}\frac{t}{\tau}}\mathbf{O} + 2\mathbf{M}e^{\lambda_\perp \frac{t}{\tau}}\mathbf{M}^T. \tag{15}$$

$$\mathbf{F} = \mathbf{O}\mathbf{\Lambda}\mathbf{O}^T + 2\mathbf{M}\lambda_\perp\mathbf{M}^T. \tag{16}$$

$$e^{\mathbf{F}\frac{t}{\tau}}\mathbf{F}^{-1}e^{\mathbf{F}\frac{t}{\tau}} - \mathbf{F}^{-1} = \tag{17}$$

$$\mathbf{O}e^{\mathbf{\Lambda}\frac{t}{\tau}}\mathbf{O}^T\mathbf{O}\mathbf{\Lambda}^{-1}\mathbf{O}^T\mathbf{O}e^{\mathbf{\Lambda}\frac{t}{\tau}}\mathbf{O}^T - \mathbf{O}\mathbf{\Lambda}^{-1}\mathbf{O}^T + \mathbf{M}(e^{\lambda_\perp \frac{t}{\tau}} - \mathbf{I_{min(N_i,N_o)}})(\lambda_\perp)^{-1}\mathbf{M}^T.$$

Where $\mathbf{M} = \frac{1}{\sqrt{2}}\begin{bmatrix} \tilde{\mathbf{V}}_\perp \\ \tilde{\mathbf{U}}_\perp \end{bmatrix}^T$. Placing these expressions into Eq. 2 gives

$$\mathbf{QQ}^T(t) = \left[\mathbf{O}e^{\mathbf{\Lambda}\frac{t}{\tau}}\mathbf{O}^T + 2\mathbf{M}e^{\lambda_\perp \frac{t}{\tau}}\mathbf{M}^T\right]\mathbf{Q}(0)$$

$$\left[\mathbf{I} + \frac{1}{2}\mathbf{Q}(0)^T\left(\mathbf{O}\left(e^{2\mathbf{\Lambda}\frac{t}{\tau}} - \mathbf{I}\right)\mathbf{\Lambda}^{-1}\mathbf{O}^T + \mathbf{M}(e^{\lambda_\perp \frac{t}{\tau}} - \mathbf{I})\lambda_\perp^{-1}\mathbf{M}^T\right)\mathbf{Q}(0)\right]^{-1}$$

$$\mathbf{Q}(0)^T\left[\mathbf{O}e^{\mathbf{\Lambda}\frac{t}{\tau}}\mathbf{O}^T + 2\mathbf{M}e^{\lambda_\perp \frac{t}{\tau}}\mathbf{M}^T\right]^T. \tag{18}$$

$$\boldsymbol{O}^T\boldsymbol{Q}(0) = \frac{1}{\sqrt{2}}\begin{pmatrix} \tilde{\mathbf{V}}(\tilde{\mathbf{G}}-\tilde{\mathbf{H}}\tilde{\mathbf{G}}) & \tilde{\mathbf{V}}(\tilde{\mathbf{G}}+\tilde{\mathbf{H}}\tilde{\mathbf{G}}) \\ \tilde{\mathbf{U}}(\tilde{\mathbf{G}}+\tilde{\mathbf{H}}\tilde{\mathbf{G}}) & -\tilde{\mathbf{U}}(\tilde{\mathbf{G}}-\tilde{\mathbf{H}}\tilde{\mathbf{G}}) \end{pmatrix}^T \begin{pmatrix} \mathbf{W}_1^T(0) \\ \mathbf{W}_2(0) \end{pmatrix}$$

$$= \frac{1}{\sqrt{2}}\begin{pmatrix} (\tilde{\mathbf{G}}-\tilde{\mathbf{H}}\tilde{\mathbf{G}})\tilde{\mathbf{V}}^T\mathbf{W}_1^T(0) + (\tilde{\mathbf{G}}+\tilde{\mathbf{H}}\tilde{\mathbf{G}})\tilde{\mathbf{U}}^T\mathbf{W}_2(0) \\ (\tilde{\mathbf{G}}+\tilde{\mathbf{H}}\tilde{\mathbf{G}})\tilde{\mathbf{V}}^T\mathbf{W}_1^T(0) - (\tilde{\mathbf{G}}-\tilde{\mathbf{H}}\tilde{\mathbf{G}})\tilde{\mathbf{U}}^T\mathbf{W}_2(0) \end{pmatrix}$$

$$= \frac{1}{\sqrt{2}}\begin{pmatrix} \mathbf{B}^T \\ -\mathbf{C}^T \end{pmatrix}, \tag{19}$$

where

$$\mathbf{B} = \mathbf{W}_2(0)^T\tilde{\mathbf{U}}(\tilde{\mathbf{G}}+\tilde{\mathbf{H}}\tilde{\mathbf{G}}) + \mathbf{W}_1(0)\tilde{\mathbf{V}}(\tilde{\mathbf{G}}-\tilde{\mathbf{H}}\tilde{\mathbf{G}}) \in \mathbb{R}^{N_h\times N_h}, \tag{20}$$

$$\mathbf{C} = \mathbf{W}_2(0)^T\tilde{\mathbf{U}}(\tilde{\mathbf{G}}-\tilde{\mathbf{H}}\tilde{\mathbf{G}}) - \mathbf{W}_1(0)\tilde{\mathbf{V}}(\tilde{\mathbf{G}}+\tilde{\mathbf{H}}\tilde{\mathbf{G}}) \in \mathbb{R}^{N_h\times N_h}. \tag{21}$$

$$\boldsymbol{O}e^{\boldsymbol{\Lambda}t/\tau} = \frac{1}{\sqrt{2}}\begin{pmatrix} \tilde{\mathbf{V}}(\tilde{\mathbf{G}}-\tilde{\mathbf{H}}\tilde{\mathbf{G}}) & \tilde{\mathbf{V}}(\tilde{\mathbf{G}}+\tilde{\mathbf{H}}\tilde{\mathbf{G}}) \\ \tilde{\mathbf{U}}(\tilde{\mathbf{G}}+\tilde{\mathbf{H}}\tilde{\mathbf{G}}) & -\tilde{\mathbf{U}}(\tilde{\mathbf{G}}-\tilde{\mathbf{H}}\tilde{\mathbf{G}}) \end{pmatrix} \begin{pmatrix} e^{\tilde{\mathbf{S}}_\lambda\frac{t}{\tau}} & 0 \\ 0 & e^{-\tilde{\mathbf{S}}_\lambda\frac{t}{\tau}} \end{pmatrix}$$

$$= \frac{1}{\sqrt{2}}\begin{pmatrix} \tilde{\mathbf{V}}(\tilde{\mathbf{G}}-\tilde{\mathbf{H}}\tilde{\mathbf{G}})e^{\tilde{\mathbf{S}}_\lambda\frac{t}{\tau}} & \tilde{\mathbf{V}}(\tilde{\mathbf{G}}+\tilde{\mathbf{H}}\tilde{\mathbf{G}})e^{-\tilde{\mathbf{S}}_\lambda\frac{t}{\tau}} \\ \tilde{\mathbf{U}}(\tilde{\mathbf{G}}+\tilde{\mathbf{H}}\tilde{\mathbf{G}})e^{\tilde{\mathbf{S}}_\lambda\frac{t}{\tau}} & -\tilde{\mathbf{U}}(\tilde{\mathbf{G}}-\tilde{\mathbf{H}}\tilde{\mathbf{G}})e^{-\tilde{\mathbf{S}}_\lambda\frac{t}{\tau}} \end{pmatrix}. \tag{22}$$

$$\boldsymbol{O}e^{\boldsymbol{\Lambda}t/\tau}\boldsymbol{O}^T\boldsymbol{Q}(0) = \frac{1}{2}\begin{pmatrix} \tilde{\mathbf{V}}(\tilde{\mathbf{G}}-\tilde{\mathbf{H}}\tilde{\mathbf{G}})e^{\tilde{\mathbf{S}}_\lambda\frac{t}{\tau}} & \tilde{\mathbf{V}}(\tilde{\mathbf{G}}+\tilde{\mathbf{H}}\tilde{\mathbf{G}})e^{-\tilde{\mathbf{S}}_\lambda\frac{t}{\tau}} \\ \tilde{\mathbf{U}}(\tilde{\mathbf{G}}+\tilde{\mathbf{H}}\tilde{\mathbf{G}})e^{\tilde{\mathbf{S}}_\lambda\frac{t}{\tau}} & -\tilde{\mathbf{U}}(\tilde{\mathbf{G}}-\tilde{\mathbf{H}}\tilde{\mathbf{G}})e^{-\tilde{\mathbf{S}}_\lambda\frac{t}{\tau}} \end{pmatrix}\begin{pmatrix} \mathbf{B}^T \\ -\mathbf{C}^T \end{pmatrix}$$

$$= \frac{1}{2}\begin{pmatrix} \tilde{\mathbf{V}}(\tilde{\mathbf{G}}-\tilde{\mathbf{H}}\tilde{\mathbf{G}})e^{\tilde{\mathbf{S}}_\lambda\frac{t}{\tau}}\mathbf{B}^T - \tilde{\mathbf{V}}(\tilde{\mathbf{G}}+\tilde{\mathbf{H}}\tilde{\mathbf{G}})e^{-\tilde{\mathbf{S}}_\lambda\frac{t}{\tau}}\mathbf{C}^T \\ \tilde{\mathbf{U}}(\tilde{\mathbf{G}}+\tilde{\mathbf{H}}\tilde{\mathbf{G}})e^{\tilde{\mathbf{S}}_\lambda\frac{t}{\tau}}\mathbf{B}^T + \tilde{\mathbf{U}}(\tilde{\mathbf{G}}-\tilde{\mathbf{H}}\tilde{\mathbf{G}})e^{-\tilde{\mathbf{S}}_\lambda\frac{t}{\tau}}\mathbf{C}^T \end{pmatrix}. \tag{23}$$

$$2\mathbf{M}e^{\boldsymbol{\lambda}_\perp\frac{t}{\tau}}\mathbf{M}^T\mathbf{Q}(0) = 2\frac{1}{\sqrt{2}}\begin{bmatrix} \tilde{\mathbf{V}}_\perp \\ \tilde{\mathbf{U}}_\perp \end{bmatrix}\begin{bmatrix} e^{\boldsymbol{\lambda}_\perp\frac{t}{\tau}} & 0 \\ 0 & e^{\boldsymbol{\lambda}_\perp\frac{t}{\tau}} \end{bmatrix}\frac{1}{\sqrt{2}}\begin{bmatrix} \tilde{\mathbf{V}}_\perp \\ \tilde{\mathbf{U}}_\perp \end{bmatrix}^T\begin{bmatrix} \mathbf{W}_1(0)^T \\ \mathbf{W}_2(0) \end{bmatrix}$$

$$= \begin{bmatrix} \tilde{\mathbf{V}}_\perp e^{\boldsymbol{\lambda}_\perp\frac{t}{\tau}}\tilde{\mathbf{V}}_\perp^T & 0 \\ 0 & \tilde{\mathbf{U}}_\perp e^{\boldsymbol{\lambda}_\perp\frac{t}{\tau}}\tilde{\mathbf{U}}_\perp^T \end{bmatrix}\begin{bmatrix} \mathbf{W}_1(0)^T \\ \mathbf{W}_2(0) \end{bmatrix}$$

$$= \begin{bmatrix} \tilde{\mathbf{V}}_\perp e^{\boldsymbol{\lambda}_\perp\frac{t}{\tau}}\tilde{\mathbf{V}}_\perp^T\mathbf{W}_1(0)^T \\ \tilde{\mathbf{U}}_\perp e^{\boldsymbol{\lambda}_\perp\frac{t}{\tau}}\tilde{\mathbf{U}}_\perp^T\mathbf{W}_2(0) \end{bmatrix}. \tag{24}$$

Putting it together we get the expressions for $\mathbf{Z}_1(t)$ and $\mathbf{Z}_2(t)$

$$\left[\boldsymbol{O}e^{\boldsymbol{\Lambda}\frac{t}{\tau}}\boldsymbol{O}^T + 2\mathbf{M}e^{\boldsymbol{\lambda}_\perp\frac{t}{\tau}}\mathbf{M}^T\right]\mathbf{Q}(0) = \tag{25}$$

$$= \frac{1}{2}\begin{pmatrix} \tilde{\mathbf{V}}(\tilde{\mathbf{G}}-\tilde{\mathbf{H}}\tilde{\mathbf{G}})e^{\tilde{\mathbf{S}}_\lambda\frac{t}{\tau}}\mathbf{B}^T - \tilde{\mathbf{V}}(\tilde{\mathbf{G}}+\tilde{\mathbf{H}}\tilde{\mathbf{G}})e^{-\tilde{\mathbf{S}}_\lambda\frac{t}{\tau}}\mathbf{C}^T \\ \tilde{\mathbf{U}}(\tilde{\mathbf{G}}+\tilde{\mathbf{H}}\tilde{\mathbf{G}})e^{\tilde{\mathbf{S}}_\lambda\frac{t}{\tau}}\mathbf{B}^T + \tilde{\mathbf{U}}(\tilde{\mathbf{G}}-\tilde{\mathbf{H}}\tilde{\mathbf{G}})e^{-\tilde{\mathbf{S}}_\lambda\frac{t}{\tau}}\mathbf{C}^T \end{pmatrix} + \begin{bmatrix} \tilde{\mathbf{V}}_\perp e^{\boldsymbol{\lambda}_\perp\frac{t}{\tau}}\tilde{\mathbf{V}}_\perp^T\mathbf{W}_1(0)^T \\ \tilde{\mathbf{U}}_\perp e^{\boldsymbol{\lambda}_\perp\frac{t}{\tau}}\tilde{\mathbf{U}}_\perp^T\mathbf{W}_2(0) \end{bmatrix}.$$

$$\mathbf{Z}_1(t) = \frac{1}{2}\tilde{\mathbf{V}}(\tilde{\mathbf{G}}-\tilde{\mathbf{H}}\tilde{\mathbf{G}})e^{\tilde{\mathbf{S}}_\lambda\frac{t}{\tau}}\mathbf{B}^T - \frac{1}{2}\tilde{\mathbf{V}}(\tilde{\mathbf{G}}+\tilde{\mathbf{H}}\tilde{\mathbf{G}})e^{-\tilde{\mathbf{S}}_\lambda\frac{t}{\tau}}\mathbf{C}^T + \tilde{\mathbf{V}}_\perp e^{\boldsymbol{\lambda}_\perp\frac{t}{\tau}}\tilde{\mathbf{V}}_\perp^T\mathbf{W}_1(0)^T. \tag{26}$$

$$\mathbf{Z}_2(t) = \frac{1}{2}\tilde{\mathbf{U}}(\tilde{\mathbf{G}}+\tilde{\mathbf{H}}\tilde{\mathbf{G}})e^{\tilde{\mathbf{S}}_\lambda\frac{t}{\tau}}\mathbf{B}^T + \frac{1}{2}\tilde{\mathbf{U}}(\tilde{\mathbf{G}}-\tilde{\mathbf{H}}\tilde{\mathbf{G}})e^{-\tilde{\mathbf{S}}_\lambda\frac{t}{\tau}}\mathbf{C}^T + \tilde{\mathbf{U}}_\perp e^{\boldsymbol{\lambda}_\perp\frac{t}{\tau}}\tilde{\mathbf{U}}_\perp^T\mathbf{W}_2(0). \tag{27}$$

We now compute the terms inside the inverse

$$\mathbf{Q}(0)^T\mathbf{M}(e^{\boldsymbol{\lambda}_\perp\frac{t}{\tau}})\boldsymbol{\lambda}_\perp^{-1}\mathbf{M}^T\mathbf{Q}(0) =$$

$$\begin{bmatrix} \mathbf{W}_1(0) & \mathbf{W}_2(0)^T \end{bmatrix}\frac{1}{\sqrt{2}}\begin{bmatrix} \tilde{\mathbf{V}}_\perp \\ \tilde{\mathbf{U}}_\perp \end{bmatrix}\begin{bmatrix} e^{\boldsymbol{\lambda}_\perp\frac{t}{\tau}} & 0 \\ 0 & e^{\boldsymbol{\lambda}_\perp\frac{t}{\tau}} \end{bmatrix}\begin{bmatrix} \boldsymbol{\lambda}_\perp & 0 \\ 0 & \boldsymbol{\lambda}_\perp \end{bmatrix}^{-1}\frac{1}{\sqrt{2}}\begin{bmatrix} \tilde{\mathbf{V}}_\perp \\ \tilde{\mathbf{U}}_\perp \end{bmatrix}^T\begin{bmatrix} \mathbf{W}_1(0)^T \\ \mathbf{W}_2(0) \end{bmatrix}$$

$$\begin{bmatrix} \mathbf{W}_1(0) & \mathbf{W}_2(0)^T \end{bmatrix}\begin{bmatrix} e^{\boldsymbol{\lambda}_\perp\frac{t}{\tau}}\boldsymbol{\lambda}_\perp^{-1}\tilde{\mathbf{V}}_\perp\tilde{\mathbf{V}}_\perp^T\mathbf{W}_1(0)^T \\ e^{\boldsymbol{\lambda}_\perp\frac{t}{\tau}}\boldsymbol{\lambda}_\perp^{-1}\tilde{\mathbf{U}}_\perp\tilde{\mathbf{U}}_\perp^T\mathbf{W}_2(0) \end{bmatrix} \tag{28}$$

$$= \left[\left(\mathbf{W}_1(0)\tilde{\mathbf{V}}_\perp e^{\boldsymbol{\lambda}_\perp\frac{t}{\tau}}\boldsymbol{\lambda}_\perp^{-1}\tilde{\mathbf{V}}_\perp^T\mathbf{W}_1(0)^T + \mathbf{W}_2(0)^T\tilde{\mathbf{U}}_\perp e^{\boldsymbol{\lambda}_\perp\frac{t}{\tau}}\boldsymbol{\lambda}_\perp^{-1}\tilde{\mathbf{U}}_\perp^T\mathbf{W}_2(0)\right)\right].$$

$$\mathbf{Q}(0)^T \mathbf{M} \lambda_\perp^{-1} \mathbf{M}^T \mathbf{Q}(0) =$$

$$2 \begin{bmatrix} \mathbf{W}_1(0) & \mathbf{W}_2(0)^T \end{bmatrix} \frac{1}{\sqrt{2}} \begin{bmatrix} \tilde{\mathbf{V}}_\perp \\ \tilde{\mathbf{U}}_\perp \end{bmatrix} \begin{bmatrix} \lambda_\perp & 0 \\ 0 & \lambda_\perp \end{bmatrix}^{-1} \frac{1}{\sqrt{2}} \begin{bmatrix} \tilde{\mathbf{V}}_\perp \\ \tilde{\mathbf{U}}_\perp \end{bmatrix}^T \begin{bmatrix} \mathbf{W}_1(0)^T \\ \mathbf{W}_2(0) \end{bmatrix}$$

$$= \begin{bmatrix} \mathbf{W}_1(0) & \mathbf{W}_2(0)^T \end{bmatrix} \begin{bmatrix} \tilde{\mathbf{V}}_\perp \\ \tilde{\mathbf{U}}_\perp \end{bmatrix} \begin{bmatrix} \lambda_\perp^{-1} \tilde{\mathbf{V}}_\perp \tilde{\mathbf{V}}_\perp^T \mathbf{W}_1(0)^T \\ \lambda_\perp^{-1} \tilde{\mathbf{U}}_\perp \tilde{\mathbf{U}}_\perp^T \mathbf{W}_2(0) \end{bmatrix}$$

$$= \begin{bmatrix} \mathbf{W}_1(0) \tilde{\mathbf{V}}_\perp \lambda_\perp^{-1} \tilde{\mathbf{V}}_\perp^T \mathbf{W}_1(0)^T + \mathbf{W}_2(0)^T \tilde{\mathbf{U}}_\perp \lambda_\perp^{-1} \tilde{\mathbf{U}}_\perp^T \mathbf{W}_2(0) \end{bmatrix}. \tag{29}$$

Now

$$\frac{1}{2} \mathbf{Q}(0)^T \mathbf{O} \left( e^{2\mathbf{\Lambda} \frac{t}{\tau}} - \mathbf{I} \right) \mathbf{\Lambda}^{-1} \mathbf{O}^T = \quad \frac{1}{4} [\mathbf{B} - \mathbf{C}] \left( e^{\mathbf{\Lambda} \frac{t}{\tau}} - \mathbf{I}_{\min(\mathbf{N_i}, \mathbf{N_o})} \right) \mathbf{\Lambda}^{-1} \begin{pmatrix} \mathbf{B}^T \\ -\mathbf{C}^T \end{pmatrix} \tag{30}$$

$$= \frac{1}{4} \left( \mathbf{B} \left( e^{2\tilde{\mathbf{S}}_\lambda \frac{t}{\tau}} - \mathbf{I}_{\min(\mathbf{N_i}, \mathbf{N_o})} \right) (\tilde{\mathbf{S}}_\lambda)^{-1} \mathbf{B}^T - \mathbf{C} \left( e^{-2\tilde{\mathbf{S}}_\lambda \frac{t}{\tau}} - \mathbf{I}_{\min(\mathbf{N_i}, \mathbf{N_o})} \right) (\tilde{\mathbf{S}}_\lambda)^{-1} \mathbf{C}^T \right). \tag{31}$$

Putting it all together

$$\mathbf{A}(t) = \mathbf{I_h} + \mathbf{B} \left( \frac{e^{2\tilde{\mathbf{S}}_\lambda \frac{t}{\tau}} - \mathbf{I}_{\min(\mathbf{N_i}, \mathbf{N_o})}}{4\tilde{\mathbf{S}}_\lambda} \right) \mathbf{B}^T - \mathbf{C} \left( \frac{e^{-2\tilde{\mathbf{S}}_\lambda \frac{t}{\tau}} - \mathbf{I}_{\min(\mathbf{N_i}, \mathbf{N_o})}}{4\tilde{\mathbf{S}}_\lambda} \right) \mathbf{C}^T$$

$$+ \mathbf{W}_2(0)^T \tilde{\mathbf{U}}_\perp \left( \frac{e^{\lambda_\perp \frac{t}{\tau}} - \mathbf{I}_{\min(\mathbf{N_i}, \mathbf{N_o})}}{\lambda_\perp} \right) \tilde{\mathbf{U}}_\perp^T \mathbf{W}_2(0) + \mathbf{W}_1(0) \tilde{\mathbf{V}}_\perp \left( \frac{e^{\lambda_\perp \frac{t}{\tau}} - \mathbf{I}_{\min(\mathbf{N_i}, \mathbf{N_o})}}{\lambda_\perp} \right) \tilde{\mathbf{V}}_\perp^T \mathbf{W}_1(0)^T. \tag{32}$$

So, final form:

$$\mathbf{Q}\mathbf{Q}^T(t) = \tag{33}$$

$$\begin{bmatrix} \left( \frac{1}{2} \tilde{\mathbf{V}}(\tilde{\mathbf{G}} - \tilde{\mathbf{H}}\tilde{\mathbf{G}}) e^{\tilde{\mathbf{S}}_\lambda \frac{t}{\tau}} \mathbf{B}^T - \frac{1}{2} \tilde{\mathbf{V}}(\tilde{\mathbf{G}} + \tilde{\mathbf{H}}\tilde{\mathbf{G}}) e^{-\tilde{\mathbf{S}}_\lambda \frac{t}{\tau}} \mathbf{C}^T + \tilde{\mathbf{V}}_\perp e^{\lambda_\perp \frac{t}{\tau}} \tilde{\mathbf{V}}_\perp^T \mathbf{W}_1(0)^T \right) \\ \left( \frac{1}{2} \tilde{\mathbf{U}}(\tilde{\mathbf{G}} + \tilde{\mathbf{H}}\tilde{\mathbf{G}}) e^{\tilde{\mathbf{S}}_\lambda \frac{t}{\tau}} \mathbf{B}^T + \frac{1}{2} \tilde{\mathbf{U}}(\tilde{\mathbf{G}} - \tilde{\mathbf{H}}\tilde{\mathbf{G}}) e^{-\tilde{\mathbf{S}}_\lambda \frac{t}{\tau}} \mathbf{C}^T + \tilde{\mathbf{U}}_\perp e^{\lambda_\perp \frac{t}{\tau}} \tilde{\mathbf{U}}_\perp^T \mathbf{W}_2(0) \right) \end{bmatrix}$$

$$\left[ \mathbf{I_h} + \frac{1}{4} \left( \mathbf{B} \left( \frac{e^{2\tilde{\mathbf{S}}_\lambda \frac{t}{\tau}} - \mathbf{I}_{\min(\mathbf{N_i}, \mathbf{N_o})}}{\tilde{\mathbf{S}}_\lambda} \right) \mathbf{B}^T - \mathbf{C} \left( \frac{e^{-2\tilde{\mathbf{S}}_\lambda \frac{t}{\tau}} - \mathbf{I}_{\min(\mathbf{N_i}, \mathbf{N_o})}}{\tilde{\mathbf{S}}_\lambda} \right) \mathbf{C}^T \right) \right.$$

$$\left. + \mathbf{W}_2(0)^T \tilde{\mathbf{U}}_\perp \left( \frac{e^{\lambda_\perp \frac{t}{\tau}} - \mathbf{I}_{\min(\mathbf{N_i}, \mathbf{N_o})}}{\lambda_\perp} \right) \tilde{\mathbf{U}}_\perp^T \mathbf{W}_2(0) + \mathbf{W}_1(0) \tilde{\mathbf{V}}_\perp \left( \frac{e^{\lambda_\perp \frac{t}{\tau}} - \mathbf{I}_{\min(\mathbf{N_i}, \mathbf{N_o})}}{\lambda_\perp} \right) \tilde{\mathbf{V}}_\perp^T \mathbf{W}_1(0)^T \right]^{-1}$$

$$\begin{bmatrix} \left( \frac{1}{2} \tilde{\mathbf{V}}(\tilde{\mathbf{G}} - \tilde{\mathbf{H}}\tilde{\mathbf{G}}) e^{\tilde{\mathbf{S}}_\lambda \frac{t}{\tau}} \mathbf{B}^T - \frac{1}{2} \tilde{\mathbf{V}}(\tilde{\mathbf{G}} + \tilde{\mathbf{H}}\tilde{\mathbf{G}}) e^{-\tilde{\mathbf{S}}_\lambda \frac{t}{\tau}} \mathbf{C}^T + \tilde{\mathbf{V}}_\perp e^{\lambda_\perp \frac{t}{\tau}} \tilde{\mathbf{V}}_\perp^T \mathbf{W}_1(0)^T \right) \\ \left( \frac{1}{2} \tilde{\mathbf{U}}(\tilde{\mathbf{G}} + \tilde{\mathbf{H}}\tilde{\mathbf{G}}) e^{\tilde{\mathbf{S}}_\lambda \frac{t}{\tau}} \mathbf{B}^T + \frac{1}{2} \tilde{\mathbf{U}}(\tilde{\mathbf{G}} - \tilde{\mathbf{H}}\tilde{\mathbf{G}}) e^{-\tilde{\mathbf{S}}_\lambda \frac{t}{\tau}} \mathbf{C}^T + \tilde{\mathbf{U}}_\perp e^{\lambda_\perp \frac{t}{\tau}} \tilde{\mathbf{U}}_\perp^T \mathbf{W}_2(0) \right) \end{bmatrix}^T.$$

$$\square$$

### A.4 Aligned dynamics

## B Implementation and simulations

### B.1 Lambda-balanced weight initialization

In practice, to initialize the network with lambda-balanced weights, we use the Algorithm bellow. In this algorithm, $\alpha$ serves as a scaling factor that controls the variance of the weights, allowing for adjustments between smaller and larger weight initializations.

### B.2 Tasks

In the following, we describe the different tasks that are used throughout the simulation studies.

**Algorithm 1** Get $\lambda$-*balanced*

---

1: **function** GET_LAMBDA_BALANCED($\lambda$, $in\_dim$, $hidden\_dim$, $out\_dim$, $\sigma = 1$)
2:     $W_1 \leftarrow \sigma \cdot$ random normal matrix($hidden\_dim, in\_dim$)
3:     $W_2 \leftarrow \sigma \cdot$ random normal matrix($out\_dim, hidden\_dim$)
4:     $[U, S, Vt] \leftarrow$ SVD($W_2 \cdot W_1$)
5:     $R \leftarrow$ random orthonormal matrix($hidden\_dim$)
6:     **for** $i = 1$ **to** $k$ **do**
7:         $s \leftarrow S[i]$
8:         $s2 \leftarrow \sqrt{\dfrac{\sqrt{\lambda^2 + 4s^2} + \lambda}{2}}$
9:         $s1 \leftarrow \sqrt{\dfrac{\sqrt{\lambda^2 + 4s^2} - \lambda}{2}}$
10:        $S_2[i,i] \leftarrow s2$
11:        $S_1[i,i] \leftarrow s1$
12:     **end for**
13:     $init\_W_2 \leftarrow U \cdot S2 \cdot R^T$
14:     $init\_W_1 \leftarrow R \cdot S1 \cdot Vt$
15:     **return** $(init\_W_1, init\_W_2)$
16: **end function**

---

### B.2.1   Random regression task

In the random regression task, the input matrix $\mathbf{X} \in \mathbb{R}^{N_i \times N}$ is drawn from a standard normal distribution, $\mathbf{X} \sim \mathcal{N}(0,1)$. The inputs are subsequently whitened to satisfy $\frac{1}{N}\mathbf{X}\mathbf{X}^\top = \mathbf{I}$. Target values $\mathbf{Y} \in \mathbb{R}^{N_o \times N}$ are generated independently from a normal distribution with variance scaled by the number of output nodes, $\mathbf{Y} \sim \mathcal{N}\left(0, \frac{1}{\sqrt{N_o}}\right)$. As $\mathbf{X}$ and $\mathbf{Y}$ are independently generated Gaussian noise, they are uncorrelated, and a linear mapping from inputs to targets does not necessarily exist.

### B.3   Figure 1

The figure illustrates three simulations conducted on the same task with varying initial $\lambda$-balanced weights respectively $\lambda = 1$, $\lambda = 0$, $\lambda = -1$. The regression task parameters were set with ($\sigma = \sqrt{0.1}$). The network architecture consisted of $N_i = 3$, $N_h = 2$, $N_o = 3$,with a learning rate of $\eta = 0.01$. The batch size is $N = 20$. The zero-balanced weights are initialized with variance $\sigma = 0.0001$. The lambda-balanced network are initialized with $sigma_{xy} = \sqrt{1}$ of a random regression task with same architecture.

### B.4   Figure 5

Panel A illustrates schematic representations of the network architectures considered: from left to right, a funnel network ($N_i = 4$, $N_h = 2$, $N_o = 2$), a square network ($N_i = 4$, $N_h = 4$, $N_o = 4$), an inverted-funnel network ($N_i = 2$, $N_h = 2$, $N_o = 4$) and a bottlenecked network ($N_i = 4$, $N_h = 2$, $N_o = 4$) . Panel B shows the Neural Tangent Kernel (NTK) distance from initialization, as defined in Fort et al. [2020], across the three architectures shown schematically. The kernel distance is calculated as:

$$S(t) = 1 - \frac{\langle K_0, K_t \rangle}{\|K_0\|_F \|K_t\|_F}.$$

The simulations conducted on the same task with eleven varying initial $\lambda$-balanced weights in $[-9, 9]$. The regression task parameters were set with ($\sigma = \sqrt{3}$). The task has batch size $N = 10$. The network has with a learning rate of $\eta = 0.009$. The lambda-balanced network are initialized with $\sigma xy = \sqrt{1}$ of a random regression task. Panel C shows the Neural Tangent Kernel (NTK) distance from initialization for the bottlenecked architectures with different architecture schematically with dimensions $N_h = 2$ and varying the dimension of $N_i = N_o = 3, 4, 5, 6$ respectively. The simulations conducted on the same task with twenty one varying initial $\lambda$-balanced weights in $[-9, 9]$. The regression task parameters were set with ($\sigma = \sqrt{3}$). The task has batch size $N = 10$. The network

has with a learning rate of $\eta = 0.009$. The lambda-balanced network are initialized with $\sigma_{xy} = \sqrt{1}$ of a random regression task.

