# OpenReview forum: "Exact Learning Dynamics of Bottlenecked and Wide Deep Linear Networks"
_NeurIPS.cc/2025/Workshop/UniReps — UniReps2025_

### Official Review · Reviewer_nrJb · 2025-09-13
**Interesting connections - clarity can be improved significantly with just a few well-placed sentences of elaboration**

**Confidence:** 4

**Review:**

Summary
- Derives exact learning dynamics for two-layer linear networks (bottlenecked) via a matrix Riccati formulation
- Essentially extends Domine & Anguita et al (2025) by making slight relaxations of assumption on hidden layer width, which has the implication of allowing analysis of the bottlenecked architecture, which is the paper's main contribution

Strengths
- Interesting finding of bottlenecked networks being in the "rich" regime regardless of lambda
- Interesting connection to LoRA architectures was briefly mentioned, but never explained. Suggestion: could use 1-2 more sentences (even if just speculation/conjecture) of how the findings / architectural motifs might inform our understanding of LoRA modules

Weaknesses
- Very limited novely/improvement from Domine & Anguita et al (2025), as the only new real piece of analysis is the bottlenecked architecture. The proofs and analysis appear almost identical to that paper.
- Poor clarity on what the task is (suggestion: either describe it briefly or be more explicit that you use the same entire problem setting from Domine & Anguita et al (2025) and clearly refer the reader there for information on the setup)
- Limited clarity in the explanation for the bottlenecked network finding. It is not clear what is meant by: "This can be attributed to their limited capacity to perfectly fit the target function. Intuitively, the networks can not learn the function in the lazy regime, since the rank of the network is smaller than that of the task.". Suggestion: This deserves slightly more elaboration to make the argument more clear.
- Typos everywhere (suggestion: do a global search of "Bottoledneck", "serpately")

**Score:**

4

**Topic Fit:**

2

---

### Official Review · Reviewer_7gc8 · 2025-09-14
**technically solid and clearly written, but seems quite incremental**

**Confidence:** 4

**Review:**

Short summary

This paper studies exact learning dynamics of 2-layer fully connected NNs where the widths of all layers are arbitrary. The authors were able to relax some the asumptions made in previous studies to yield a more general result.



Strengths

1. The paper is clearly presented in all its parts.

2. The authors clearly show how they relax assumptions made in earlier work.

3. The relation to rich and lazy regimes is interesting and illuminating.

4. The technical results appear to be sound

Weaknesses

1. I found the paper to be a bit over-selling, e.g. when mentioning “parallels between brain circuitry and machine learning architectures”. Even the word “deep” in the title could be misleading, as the derivation does not easily extend to more the two layers.

2. Some refs are missing, e.g in L51: “initialization variance and layer-wise learning rates must scale in the infinite-width limit to preserve stable activations, gradients, and output” - citing the $\mu$P line of work by Yang et al. and Bordelon et al. seems in order.

3. I felt that some of the defs are not well motivated, e.g. \lambda-balance and I would add the definition of kernel distance.

Overall, this contribution is technically solid and clearly written, but seems quite incremental wrt previous work.

**Score:**

3

**Topic Fit:**

3

---

### Official Review · Reviewer_aetc · 2025-09-15
**Exact Learning Dynamics of Bottlenecked and Wide Deep Linear Networks: review**

**Confidence:** 4

**Review:**

This long abstract presents new results on two-layered linear networks and their learning dynamics; in particular the authors show an exact solution for the matrix Riccati differential equation that governs these learning dynamics, under less restrictive constraints with respect to the existing literature.
The work is interesting and theoretically sound, and the sections are well-written and organized; moreover, the experiments match the theoretical results. I would recommend acceptance at the workshop, but I would like to raise just some minor issues:
- in the abstract it is mentioned the potential to advance the understanding of cerebellum-like structures or LoRA adaptations, but in the manuscript this part is mentioned only in the conclusions; I would remove it for the sake of space, to include more relevant things;
- What is the meaning of "whitened input" at line 91? \tilde{\Sigma}^{xx} has not been defined;
- although I believe it's a matter of space, the main result has not been included in the main text; I believe that adding it would make the long abstract more complete;
- throughout the whole manuscript, the authors wrote "bottolednecked" while it would be correct to write "bottlenecked";
- even if it is a known concept in the related literature, I would define (even quickly) what are the lazy/rich regimes
- it's difficult to interpret the results for the NTK in Figure 1

**Score:**

4

**Topic Fit:**

3

---

### Official Review · Reviewer_Pabt · 2025-09-16
**Extension of analytical methods for novel architectures**

**Confidence:** 3

**Review:**

**Summary:**

This paper describes the application of Riccati matrix framework for solving ODEs to different neural network architectures. Rather than giving solutions directly in terms of the weight space, this formalism instead describes the evolution of a set of overlap parameters given by $QQ^T(t)$ with $Q = [W_1 \; W_2^T]$ . The authors describe using a weakened set of assumptions compared with previous work that allow them to give exact solutions for a wider set of architectures in which bottlenecks play a key role.

**Strengths:**

*Quality:*

- The authors give exact solutions for a wider class of networks, and plot the evolution of loss and parameters side-by-side with analytical solutions

*Clarity:*

- The main parts of the paper are exceedingly well-written and I think they present a number of difficult key concepts in an accessible manner; this alone would arguably be a good reason for acceptance

*Significance:*

- Extending the classes of neural networks for which we have exact solutions is good
- The authors state that they are working on applying these results to extending training regimes such as LoRA; not a lot of information is given about how this would work so it is too early to judge, but if possible it would be a significant step in making exact solutions of linear networks relevant to modern machine learning practice

**Weaknesses:**

*Quality:*

- In Fig 2B/C, are these simulations of the evolution from NTK? How do these correspond to the analytical solutions? If possible, it would be nice to plot the analytical solutions here as well.
- Barely any information about the math itself is given in the main text, which stays at a qualitative/conceptual level, apart from a note that they are following Domine et al (2024). In contrast, in the appendix, the derivations are given without a lot of additional information about how they work. In particular, they don't really explain what the key differences or innovations in their technical/analytical method compared with prior work, which makes it difficult to assess how novel/significant the methodological advances are without a high level of familiarity with prior work

*Clarity:*

- There are a number of typos that should be corrected before full publication (such as "bottolednecked" for bottlenecked)

*Significance:*

- I expect that this is intended for follow-up work, but my sense is that the link to LoRA may be more complicated than what the authors state in this paper; it would be interesting to have a bit more information about how this develops

**Score:**

3

**Topic Fit:**

3